# Transcriptomic and Metabolomic Insights into Benzylisoquinoline Alkaloid Biosynthesis in Goldthread (*Coptis trifolia*)

**DOI:** 10.3390/ijms26199704

**Published:** 2025-10-05

**Authors:** Yoo-Shin Koh, Fanchao Zhu, Yoojeong Hwang, Mi-Jeong Yoo

**Affiliations:** 1Department of Psychology, Duke University, Durham, NC 27708, USA; yoo-shin.koh@duke.edu; 2Interdisciplinary Center for Biotechnology, University of Florida, Gainesville, FL 32608, USA; fzhu9482@ufl.edu; 3Department of Otolaryngology, University of Florida, Gainesville, FL 32610, USA; 4Department of Biology, Clarkson University, Potsdam, NY 13699, USA; hwangy.2023@gmail.com

**Keywords:** *Coptis trifolia*, benzylisoquinoline alkaloid biosynthesis, BIA, transcriptomics, de novo transcriptome assembly

## Abstract

*Coptis trifolia* (threeleaf goldthread) offers a valuable comparative system for investigating the evolution and regulation of benzylisoquinoline alkaloid (BIA) synthesis. In this study, we analyzed the leaf and root transcriptomes of *C. trifolia* using both long-read and short-read RNA-Sequencing. We assembled 41,926 unigenes (≥500 bp) and identified 37 genes related to BIA biosynthesis, including two transcription factors, *bHLH1* and *WRKY1*. The number of BIA genes identified in *C. trifolia* was comparable to that in other *Coptis* species. Transcriptome analysis revealed that most of these genes were more highly expressed in roots than leaves. Consistent with previous studies, *C. trifolia* contained a single (*S*)-stylopine synthase (*SPS*) gene homolog, potentially multifunctional for (*S*)-canadine synthase (*CAS*), (*S*)-cheilanthifoline synthase (*CFS*), and *SPS*. Transcriptome and untargeted metabolomic data indicated greater variation in root samples than leaf samples, although slightly more differentially expressed transcripts and metabolites were observed in leaves. Targeted metabolite profiling showed higher BIA accumulation in roots, with epiberberine being the most abundant, followed by coptisine, berberine, and columbamine. These results provide essential genomic resources for comparative analysis of the BIA pathway across Ranunculaceae, targeted gene function studies for metabolic bioengineering, and conservation strategies for *C. trifolia*, a member of an early-diverging clade within the genus with limited genetic resources.

## 1. Introduction

Plants produce a diverse array of compounds, commonly known as phytochemicals, that help them cope with abiotic and biotic stresses and have been used to treat various diseases. With the increase in global health concerns such as antibiotic resistance, chronic inflammation, and metabolic disorders, there is increasing interest in studying the therapeutic properties of these natural compounds [1,2]. Among the most potent of these compounds are benzylisoquinoline alkaloids (BIAs), which include well-known herbal medicines such as morphine from *Papaver somniferum* L. [1]. Many traditional medicines, such as Traditional Chinese Medicine (TCM), use BIAs like berberine, a major alkaloid in *Coptis* species that has been used for various conditions and is now studied for its antidiabetic and lipid-lowering effects [3,4]. As a result, *Coptis* stands out as a strong candidate, as many other bioactive ingredients, such as palmatine and coptisine, are derived from *Coptis* species, demonstrating significant antibacterial, anti-inflammatory, and hepatoprotective properties [3,5]. Despite the importance of these compounds, the biochemical pathways responsible for their production remain only partially understood.

To facilitate investigation into these BIA biosynthetic pathways, researchers have increasingly prioritized *Coptis* species with well-documented metabolite profiles and growing genomic resources. Among them, *C. chinensis*, *C. deltoidea* C.Y. Cheng et Hsiao, and *C. teeta* Wall, collectively known as “Huang-lian”, have been staples of TCM for over two millennia, used to treat dysentery, fever, and various infections [6]. These species produce a chemically diverse suite of BIAs, including berberine, palmatine, jatrorrhizine, epiberberine, and coptisine, compounds with well-documented antibacterial, anti-inflammatory, hepatoprotective, and antitumor effects [3]. *Coptis chinensis* has also been incorporated into functional foods such as herbal teas and capsules aimed at improving glycemic control and reducing oxidative stress, reflecting its antioxidative and hypoglycemic properties [4,7]. Recent advances in genomics and transcriptomics have enabled the identification of 52 to 64 candidate genes involved in BIA biosynthesis in *C. chinensis*, *C. deltoidea*, and *C. teeta*, marking important progress in unraveling these complex pathways [8,9,10,11].

BIA originates from L-tyrosine. Dopamine and 4-hydroxyphenylacetaldehyde condense via norcoclaurine synthase (NCS) to yield (*S*)-norcoclaurine, the first pathway-specific intermediate [9,10,11]. Through the canonical sequence, (*S*)-norcoclaurine is converted to the key intermediate (*S*)-reticuline, with (*S*)-coclaurine as an intermediate and branch point. Then, (*S*)-reticuline is converted to (*S*)-scoulerine by the berberine bridge enzyme (BBE), followed by CYP719A-mediated steps, O-methyltransferases, and final oxidation, which produces epiberberine, berberine, coptisine, and columbamine [9,10,11].

Yet significant gaps remain: key steps in the coptisine biosynthetic pathway remain unclear, as the genes encoding (*S*)-cheilanthifoline synthase (*CFS*) and (*S*)-stylopine synthase (*SPS*) have not yet been fully identified or functionally validated. These enzymes play critical roles in the mid-to-late stages of the BIA pathway by catalyzing oxidative steps that ultimately lead to the formation of coptisine, a major bioactive alkaloid in *Coptis* species. The issue is compounded by the fact that BIA biosynthesis is not uniformly active across tissues or developmental stages; rather, it is subject to tight spatial, temporal, and transcriptional regulation [12]. Enzyme expression often varies between roots and leaves, and biosynthetic intermediates are known to be compartmentalized within specific cell types or subcellular organelles. These layers of regulation pose a major hurdle for efforts to reconstruct BIA pathways in heterologous systems. While recent advances in synthetic biology have allowed partial reconstitution of BIA biosynthesis in engineered yeast and bacterial platforms [13,14], these systems frequently fail to replicate the nuanced regulatory environment and compartmental dynamics of native plant tissues. As a result, they often produce only early or simplified pathway intermediates, rather than complex end-products such as coptisine. To complete our understanding of BIA biosynthesis, functional studies in native systems such as *Coptis* remain essential.

*Coptis trifolia* (L.) Salisb., or threeleaf goldthread, offers a valuable comparative system for investigating the evolution and regulation of BIA biosynthesis. This rhizomatous perennial herb inhabits cool, acidic wetlands across eastern North America and parts of Eurasia, where it forms endomycorrhizal associations and occupies ecological niches distinct from its Asian congeners [15]. Phylogenetically, *C. trifolia* belongs to an early-diverging clade within the genus, providing an evolutionary framework for exploring pathway diversification [16,17,18]. Phytochemical analyses indicate that *C. trifolia* accumulates high levels of coptisine and berberine, but lacks palmatine and hydrastine, distinguishing it from both *C. chinensis* and *Hydrastis canadensis* L. [5]. Despite historical medicinal use by Native American communities for treating infections, oral conditions, and digestive ailments, *C. trifolia* remains largely uncharacterized at the molecular level.

Given its distinct evolutionary lineage and unique metabolite profile, *C. trifolia* represents a promising system for identifying previously undetected genes and regulatory elements involved in BIA biosynthesis. Studying *C. trifolia* provides a unique opportunity to fill in key gaps in the BIA pathway and examine the influence of evolutionary divergence and ecological adaptation on the biosynthesis of specialized metabolites. Insights from this system could enhance metabolic engineering initiatives, guide conservation measures for endangered medicinal plants, and uncover novel genetic tools for plant-based drug discovery.

Thus, this study aims to: (1) investigate the leaf and root transcriptomes of *C. trifolia* using both long-read and short-read RNA sequencing (RNA-Seq), (2) identify genes involved in BIA biosynthetic pathways, (3) characterize metabolite profiles in both tissues using an untargeted approach, and (4) quantify the contents of specific alkaloids in leaf and root samples. Hence, this study deepens our understanding of the molecular basis of BIA biosynthesis and phytochemical diversity within *Coptis* species and provides valuable genetic resources for future research on *C. trifolia* populations.

## 2. Results

### 2.1. Transcriptome Assembly

To characterize the leaf and root transcriptomes of *C. trifolia*, a total of 5,768,718 long reads and 814,203,228 short reads were obtained after quality check (Appendix A). Oxford Nanopore Technologies (ONT) long-read sequencing data resulted in 58,103 and 55,191 transcripts from leaf and root samples, respectively, which merged into 86,344 transcripts. The further filtering based on the expression levels of transcripts based on short reads (Transcripts Per Million (TPM) ≥ 2) reduced to 41,280 transcripts (Figure 1).

The leaf and root transcriptomes shared 24,433 transcripts (59.2%), while 7513 and 9334 were unique in the leaf and root samples, respectively (Appendix A). Illumina short reads were assembled into 66,992 and 65,798 transcripts from leaf and root samples, respectively, which merged into 88,010 transcripts. These transcripts were reduced to 33,598 based on the expression levels (TPM ≥ 2), with 14,423 transcripts common in both tissues, while 16,760 and 2415 transcripts were exclusive to leaf and root samples, respectively (Appendix A). The two assemblies generated from long reads and short reads were refined by extracting the longest coding sequence (CDS) per transcript. Then, 25,172 and 26,655 transcripts (Figure 1) were clustered using CD-HIT v4.6.8 at 95% sequence similarity and compared with *C. chinensis* protein database (40,011 proteins) using BLASTp v2.15.0. The final reference transcriptome comprises 41,926 transcripts, 94% of which exhibited significant similarity to *C. chinensis* proteins based on BLASTp analysis (E-value < 1 × 10^−10^).

### 2.2. Transcriptome Analysis Using RNA-Seq Data

To identify differentially expressed (DE) transcripts between leaf and root tissues, an average of 135 million clean short reads per replicate were mapped to the reference transcriptome generated above. On average, 93.7% of reads successfully aligned to the transcriptome (Appendix A). Overall transcriptome profiling revealed greater variation among root samples compared to leaf samples (Figure 2A). Among transcripts with TPM ≥ 2, 11,715 (27.9%) and 10,450 (24.9%) transcripts were expressed in leaf and root tissues, respectively, with 6607 transcripts shared between both tissues. Comparative analysis between the two tissues identified 4781 and 2363 DE transcripts in leaf and root tissues, respectively (Figure 2B).

Functional enrichment analysis showed that transcripts upregulated in leaves were predominantly associated with “photosynthesis,” while those upregulated in roots were enriched for “response to stimuli” and “secondary metabolic process” (Appendix A). In addition, DE transcription factors showed enrichment for “circadian rhythm” and “response to light stimulus” in leaves, while “regulation of transcription, DNA-templated” and “regulation in secondary metabolic process” were enriched in roots (Appendix A). These findings align with the primary physiological roles of leaf and root tissues: leaves act as the main site of photosynthesis to support plant growth and energy production, while roots often confront diverse environmental stressors and accumulate specialized secondary metabolites for defense and adaptation.

### 2.3. Identification of Full-Length Transcripts Involved in BIA Biosynthesis

To identify genes involved in BIA biosynthesis, known BIA biosynthetic genes were first retrieved from NCBI and UniProt databases. Homologous sequences were then identified from the reference transcriptome using BLASTp and subsequently analyzed through phylogenetic comparison. As a result, 37 genes related to BIA biosynthesis were identified along with two transcription factors, *bHLH1* and *WRKY1*, known for BIA synthesis [19,20,21] (Appendix A). The gene numbers identified from transcriptomes of *C. trifolia* are similar to those from other *Coptis* species (Appendix A). For example, there are five *NCS*, two (*S*)-norcoclaurine 6-O-methyltransferase (*6OMT*), one 3′-hydroxy-N-methyl-(*S*)-coclaurine 4′-O-methyltransferase (*4′OMT*), two (*R*,*S*)-reticuline 7-O-methyltransferase (*7OMT*), and four columbamine O-methyltransferase (*CoOMT*) genes (Appendix A).

However, two genes showed differences in the numbers. There are two to five (*C. trifolia*) or seven (*C. chinensis*) (*S*)-coclaurine N-methyltransferase (*CNMT*) genes (Appendix A). Phylogenetic analysis revealed two distinct groups of *CNMT*, with the *CNMT*-I clade comprising two subclades: CNMT1 and CNMT2. CNMT1 includes sequences from both *Coptis* and other Ranunculaceae species, whereas CNMT2 is composed exclusively of *Coptis* sequences. (Appendix A). Higher expression of both CNMT homologs of *C. trifolia* in the CNMT-I clade in roots and the previous functional study on *C. japonica* [22] suggests that the CNMT1 clade contains putative CNMT, leading to the identification of two CNMT homologs in *C. trifolia* and *C. chinensis* (Appendix A). Therefore, *Ctr_CNMT1* may represent a functional CNMT, a hypothesis further supported by its highest expression level, although this inference should be tested through functional analysis.

Another gene family, the cytochrome P450 members, also displayed differences in gene numbers. Three genes, (*S*)-corytuberine synthase (*CTS*, CYP80G), (*S*)-N-methylcoclaurine-3′-hydroxylase (*NMCH*, CYP80B), and (*S*)-canadine synthase (*CAS*, CYP719A), contain a cytochrome P450 domain. All four *Coptis* species possess only one copy of *CTS* and *NMCH* (Appendix A). In contrast, multiple homologs of *CAS* were identified and formed groups based on their phylogenetic position (Figure 3).

*CAS* homologs in the Papaveraceae formed two distinct clades of *CFS* and *SPS* with the *SPS* clade containing genes known to possess both CAS and SPS activities [23,24]. However, *CAS* homologs from the Berberidaceae and Ranunculaceae exclusively clustered into a single *CAS* clade (Figure 3). Interestingly, *Coptis* homologs were found in two separate clades, Coptis-I and Coptis-II. Functional studies have shown that most genes in the *Coptis*-I CAS lack CAS/CFS/SPS activities [11,25]. Conversely, *Coptis* genes in the *Coptis*-II clade are suggested, based on previous functional studies, to exhibit multifunctional enzyme activities, including CAS, CFS, SPS, and (*S*)-nandinine synthase (NAS) activities [11,26,27] (Figure 3), although additional validation is required. A homolog from *Xanthorhiza simplicissima* Marshall, *Xanthorhiza.CYP719.CFS/NAS/SPS*, was shown to possess CFS, NAS, and SPS activities [27], supporting the hypothesis that multifunctionality in *CAS* genes may have evolved in the Coptideae lineage. However, further functional characterization of other Ranunculaceae members is needed to test this hypothesis.

Most of the identified genes related to BIA biosynthesis (Figure 4A) exhibited higher expression in root tissue compared to leaf tissue, except for *NCS1c*, which showed higher expression in leaves (Figure 4B). Among the three *CAS* homologs, *SPS* exhibited notably higher expression levels than *CAS1* and *CAS2* in the roots of *C. trifolia* (Figure 4B) and *C. deltoidea* (Appendix A; [9]). These results are consistent with the functional characterization of *CAS* homologs in *C. chinensis*, in which only *SPS* (IFM89_021655) was shown to possess CAS/NAS/SPS activities [11].

In addition to structural genes, two transcription factors, *bHLH1* and *WRKY1*, previously reported to regulate BIA biosynthesis in *Coptis* species [19,20,21], also exhibited higher expression in roots than leaves (Figure 4B).

### 2.4. qRT-PCR Validation of RNA-Seq Results for 15 BIA Biosynthetic Genes

To validate the RNA-Seq results, we performed qRT-PCR on 15 genes involved in BIA biosynthesis. As shown in Appendix A, the qRT-PCR results showed expression patterns generally consistent with the RNA-Seq data (*r*^2^ = 0.5781), supporting the reliability of the transcriptome analysis. However, a few genes, such as *SPS*, exhibited notably lower expression, while others, including *BBE* and (S)-tetrahydroprotoberberine oxidase (*STOX*), showed higher expression in the qRT-PCR results compared to RNA-Seq. These discrepancies may be due to the low expression levels of these genes in leaf tissue, which can hinder efficient amplification during the early PCR cycles, leading to greater variability in the qRT-PCR results (Figure 5).

### 2.5. Untargeted and Targeted Metabolite Profiling

Overall metabolite profiling was investigated and compared between leaf and root tissues using a UHPLC-tims-TOF MS system. As a result, a total of 4002 metabolites were detected and quantified. Overall variance in the dataset and clustering pattern was similar to transcriptomic data (Figure 2A) with greater variation in root samples compared to leaf samples (Figure 6A). A similar result was obtained from Partial Least Squares Discriminant Analysis (PLS-DA). Among 4002 metabolites (both compounds and features), 2226 were differentially accumulated between leaf and root tissues, with 978 and 1248 metabolites in leaf and root tissues, respectively (Figure 6B). Metabolites that accumulated more abundantly in leaf tissues were enriched in “phenylalanine, tyrosine, and tryptophan biosynthesis pathways”, while those more abundant in root tissues were enriched in “nicotinate and nicotinamide metabolism” as well as “tyrosine metabolism” (Appendix A). Tyrosine, a precursor of BIAs that accumulates in both leaf and root tissues, shows enrichment in its biosynthetic pathway, consistent with its roles in BIA production. In addition, the three aromatic amino acids, phenylalanine, tyrosine, and tryptophan, serve as precursors for a variety of specialized metabolites, such as pigments, alkaloids, and hormones, many of which are synthesized or stored in leaves, supporting their enrichment in leaf tissue. In contrast, the enrichment of “nicotinate and nicotinamide metabolism” in root tissues aligns with their known involvement in plant stress responses, which is consistent with the transcriptomics data observed earlier.

Among BIAs, only berberine was detected in untargeted profiling, which exhibited higher accumulation in root samples. Thus, we investigated six BIAs using the targeted method. Six metabolites were targeted and analyzed, and all of them showed higher accumulation in root samples than leaf samples (Figure 7). Epiberberine exhibited the highest abundance, followed by coptisine, berberine, and columbamine. Jatrorrhizine and palmatine were accumulated relatively less than other BIA compounds. Notably, palmatine was detected in *C. trifolia*, in contrast to a previous report by Kamath et al. [5], which indicated its absence.

## 3. Discussion

*Coptis* species produce a variety of phytochemicals, particularly BIAs, which have long been used for medicinal purposes. Thus, several transcriptomic studies have been conducted to elucidate BIA biosynthetic pathways, with a particular focus on candidate genes involved in BIA biosynthesis in *C. chinensis*, *C. deltoidea*, and *C. teeta* [8,9,10,11]. As a member of a sister clade to these three medicinal species, *C. trifolia* presents a valuable comparative system for investigating the evolution and regulation of BIA synthesis. Although the genome of *C. trifolia* is not yet available, transcriptome profiling via RNA-seq offers an effective approach for identifying genes involved in BIA biosynthesis.

### 3.1. High-Quality Coptis trifolia Transcriptome Assembly via Integrated ONT and Illumina Sequencing

We leveraged the complementary strengths of ONT and Illumina sequencing to assemble a robust leaf and root transcriptome for *C. trifolia*. ONT sequencing provides long reads capable of capturing full-length transcripts, but its relatively high error rate can pose challenge for accurate transcript reconstruction. In contrast, Illumina short read sequencing offers high coverage and low error rate, providing a valuable complement to ONT data. Therefore, in this study, we analyzed the leaf and root transcriptomes of *C. trifolia* using both ONT and Illumina sequencing to leverage the strengths of each technology.

After the initial assembly, we applied filtering steps to enhance transcriptome quality. All transcripts with extremely low expression (TPM < 2 in either leaf or root samples) were removed, since such weakly expressed sequences are often assembly artifacts or noise. Indeed, transcripts with low expression levels tend to have questionable biological significance, so excluding them enriches the dataset for truly expressed genes. Additionally, to minimize redundancy, we retained only the longest CDS for each gene locus, discarding shorter isoforms. This representative longest-isoform approach is a common practice to reduce transcriptome complexity and ensure that each gene is represented by a single, most complete transcript [28]. Through these quality-focused measures, the total number of assembled transcripts was reduced from approximately 88,000 to 41,926 in the final transcriptome. Importantly, this reduction reflects the elimination of low-confidence and redundant sequences rather than loss of meaningful data. The result is a more concise and high-confidence reference transcriptome enriched for well-supported transcripts.

We evaluated the completeness and accuracy of the refined transcriptome by BLAST v2.15.0 validation against well-characterized reference proteomes. Each *C. trifolia* transcript was compared to known protein sequences from *Arabidopsis thaliana* (a model plant) and *C. chinensis* using a stringent significance threshold (e.g., BLAST E-value < 1 × 10^−10^). The majority of transcripts showed strong homology to proteins in these databases. For example, approximately 93% of the transcripts matched a protein in the *A. thaliana* proteome, and about 94% had a hit in the *C. chinensis* proteome, meeting the significance cutoff. This high proportion of matches indicates that nearly all transcripts in our assembly correspond to bona fide protein-coding genes found in well-studied plants. The BLAST-based validation thus confirms that our filtering and refinement steps improved overall assembly quality, yielding a reliable reference transcriptome suitable for downstream functional analyses.

### 3.2. Differential Gene Expression Between Leaves and Roots

Using the high-quality reference transcriptome, we examined differential gene expressions between leaf and root tissues of *C. trifolia*. We identified 7144 transcripts as significantly differentially expressed between the two organs under stringent criteria (false discovery rate (FDR) < 0.05 and fold-change ≥ 4). This set of DE transcripts constitutes roughly 17.0% of the reference transcripts. Functional annotation of the DE transcripts revealed patterns consistent with the distinct physiological roles of leaves and roots. Transcripts up-regulated in leaves were strongly enriched for photosynthesis-related genes, reflecting the leaf’s primary role in light energy capture. In contrast, transcripts more highly expressed in roots were predominantly associated with secondary metabolite biosynthesis and transport, which aligns with the root’s role in producing and accumulating medicinal compounds (Appendix A). These findings match expectations for a medicinal plant: photosynthetic functions dominate in the foliage, whereas biosynthetic pathways for specialized metabolites are active in the root.

Our results are in line with observations from a related species. In a comparable full-length transcriptome study of *C. deltoidea*, which utilized PacBio single-molecule long-read sequencing instead of ONT, Zhong et al. [9] reported a similar scale of differential gene expression. Specifically, approximately 34% of *C. deltoidea* transcripts (25,391 out of 75,438) were differentially expressed between organs, based on FDR < 0.05 and a fold-change ≥ 2. In our *C. trifolia* dataset, we observed a comparable proportion of transcripts showing significant differential expression when we applied the same criteria (35.9%; 8689 out of 41,926 transcripts). This similarity in DE transcript proportion suggests that our integrative ONT/Illumina approach achieved a sensitivity for detecting expression differences on par with the PacBio-based study. Together with the functional enrichment results, this comparison supports the reliability and thoroughness of our transcriptome assembly and expression analysis pipeline.

### 3.3. Accumulation of BIA and Their Biosynthetic Genes in C. trifolia

BIA represents one of the most important classes of plant secondary metabolites, and *Coptis* species are a key plant group known for producing a diverse array of these compounds. Previous studies on three *Coptis* species, *C. chinensis*, *C. deltoidea*, and *C. teeta*, have investigated BIA contents and identified genes involved in BIA biosynthesis [8,9,10,11]. However, no comprehensive study has been conducted on *C. trifolia* that can offer valuable potential for comparative research on the evolution and regulation of BIA synthesis. To address this gap, along with the transcriptome analysis, we examined metabolite profiling of leaf and root tissues of *C. trifolia* using both untargeted and targeted methods.

Consistent with the transcriptome profiling results, leaf and root samples of *C. trifolia* were clearly separated based on untargeted metabolite profiles (Figure 6). Functional enrichment analysis revealed that metabolites more abundant in leaf tissue were associated with the “phenylalanine, tyrosine, and tryptophan biosynthesis” pathway, whereas those enriched in root tissue were involved in “nicotinate and nicotinamide metabolism” and “tyrosine metabolism” (Appendix A). As the aromatic amino acids, phenylalanine, tyrosine, and tryptophan are interconnected through the shikimate pathway, phenylalanine and tryptophan serve as key precursors for various plant secondary metabolites, including flavonoids and phytohormones. Given the metabolic role of leaf tissue, the enrichment of this pathway in leaves is consistent with its involvement in photosynthesis-related and biosynthetic activities. Additionally, tyrosine serves as a direct precursor of BIAs, further supporting its relevance in both tissues. In root tissue, the enrichment of “nicotinate and nicotinamide metabolism”, a pathway involved in plant development and stress response [29], aligns with transcriptomic findings, which showed that DE transcripts in roots were significantly enriched for the Gene Ontology category “response to stimuli.” Notably, “tyrosine metabolism” was enriched in both tissues (Appendix A), consistent with the central role of tyrosine in BIA biosynthesis (Figure 4). This is further supported by the absence of significant differential expression in genes associated with the tyrosine biosynthetic pathway, suggesting shared regulation or balanced metabolic flux between tissues.

Among the six BIA metabolites examined, *C. trifolia* exhibited higher or comparable levels of epiberberine and columbamine relative to other *Coptis* species [9,18,30]. Notably, epiberberine is rarely produced, columbamine typically accumulates at very low levels (<4 mg/g) in most *Coptis* species, with the exception of *C. chinensis.* Therefore, the relatively high concentrations of both epiberberine and columbamine in *C. trifolia* suggest that this species may serve as a valuable alternative source of these bioactive compounds.

Genes involved in BIA biosynthesis have been identified from transcriptomics data of *C. chinensis* [8,20,26,31], *C. japonica* [19,21], *C. deltoidea* [9], and *C. teeta* [8], as well as from the genomic data of *C. chinensis* [10,11]. Functional characterization of selected genes from *C. chinensis* [11,26] and *C. japonica* [22,25,32] has been performed through heterologous expression in yeast. In our study, putative BIA biosynthetic genes were identified in *C. trifolia* using BLAST and phylogenetic analyses (Appendix A). The number of gene copies in *C. trifolia* was comparable to those reported in the genome of *C. chinensis*, while some previous studies reported much higher number of genes, which may reflect isoforms of the same gene and warrants further examination (Appendix A). Notably, *C. chinensis* exhibits tandem duplication in certain genes [11]. For example, while *C. japonica* and *C. trifolia* each contain two copies of the *CAS* gene, *C. chinensis* possesses four, all of which fall within the *Coptis*-I clade (Figure 3).

Interestingly, unlike members of Papaveraceae, which possess two distinct clades corresponding to *CFS* and *SPS*, members of Ranunculaceae form a single *CAS* clade, with *Coptis* species further diverging into two separate subclades (Figure 3). Based on functional studies in *C. chinensis* [11] and *C. japonica* [27], as well as the low expression of this gene in *C. trifolia* (Figure 4B) and *C. deltoidea* (Appendix A), it is unlikely that members of this clade confer CFS activity. However, Wu et al. [26] reported that *CcCYP719A2* from *C. chinensis* may exhibit CFS activity. Instead, *CAS* homologs within the *Coptis*-II clade appear to carry multifunctional enzymatic roles, including CFS, CAS, NAS, and SPS activities [11,26,27] (Figure 3). Supporting this inference, *Ctr_SPS* and *Cdel_SPS* showed significantly higher expression in roots compared to leaves (Figure 4B and Figure 5). These findings suggest lineage-specific functional divergence of *CAS* genes. In Papaveraceae, two whole-genome duplication (WGD) events likely led to the evolution of two distinct genes, *CFS* and *SPS*/*CAS* [11,33]. In contrast, Ranunculaceae appears to retain a multifunctional *CAS* gene capable of CAS/CFS/NAS/SPS functions, as exemplified in *Coptis* lineage. Although the *Coptis* lineage has experienced a single round of WGD [11] and possesses additional *CAS* homologs, the multifunctionality remains consolidated within a single clade (Figure 3).

Genes related to BIA exhibited high expression in roots than leaves (Figure 4B). Notably, two genes, *STOX* and *SPS*, were highly expressed in roots, in agreement with the higher accumulation of epiberberine, berberine, columbamine, and coptisine in roots (Figure 6). These findings support the reliability and utility of our reference transcriptome for inferring BIA biosynthetic genes in *C. trifolia*. In addition to structural genes, we identified two transcription factors that may regulate BIA biosynthesis, as previously reported in other *Coptis* species. DE transcription factors were enriched for “Circadian rhythm” and “Response to light stimulus” in leaves, while “Regulation of transcription, DNA-templated” and “regulation in secondary metabolic process” were enriched in roots (Appendix A). Previous studies have shown that *bHLH1* and *WRKY1* may regulate BIA biosynthesis by interacting with structural genes. Specifically, *bHLH1* interacts with *BBE* and *SPS* in *C. chinensis* [20] and *4′OMT* and *SPS* in *C. japonica* [19], while *WRKY1* interacts with *NMCH* in *C. japonica* [21]. In *C. trifolia*, we identified homologs of *bHLH1* and *WRKY1* [19,20,21], both of which were differentially expressed between roots and leaves (Figure 3B). Based on expression levels, the *bHLH1* homolog may exert a more significant regulatory influence in BIA biosynthesis than *WRKY1*, as *WRKY1* expression was low in both tissues (TPM < 2). Although these two transcription factors directly activate multiple BIA biosynthetic genes via specific DNA binding motifs in *C. chinensis* [20] and *C. japonica* [19,21], future work, such as leveraging orthologous promoters, motif scans, and functional assays, will be essential to map direct targets and validate this network in *C. trifolia*.

## 4. Materials and Methods

### 4.1. Plant Materials

Wild *C. trifolia* plants were collected from Adirondack Park in Wanakena, St. Lawrence County, New York, USA, and cultivated in the greenhouse at Clarkson University. One year after transplantation, we tagged newly emerged leaves from rhizome. Then, mature leaf with its roots, including rhizome, was collected for each biological replicate as shown in Appendix A. Two individuals with similar morphology were included in each replicate, as the age of this perennial species cannot be determined. For transcriptomic and metabolomic analyses, three and five biological replicates were prepared, respectively. All samples were immediately frozen in liquid nitrogen and stored at −80 °C until extraction.

### 4.2. RNA Extraction, Library Construction, and RNA-Sequencing

Total RNAs were extracted from triplicate leaf and root samples using the RNeasy Plant Mini Kit (Qiagen, Valencia, CA, USA). RNA quality and quantity were assessed using a NanoDrop One C Microvolume UV-Vis Spectrophotometer (Thermo Fisher Scientific, Waltham, MA, USA) and an Agilent 2100 Bioanalyzer (Agilent, Santa Clara, CA, USA). The extracted RNAs were processed for two complementary RNA sequencing approaches. For ONT sequencing, equal amounts of RNA from three leaf and three root samples were pooled and submitted to the NextGen DNA Sequencing core facility at the University of Florida (UF), where sequencing was performed using the PromethION platform. For Illumina RNA-Seq, individual RNA samples were used to construct libraries with the NEBNext UltraExpress^®^ RNA Library Prep Kit and unique-dual indexes, following the manufacturer’s protocol (New England Biolabs, Ipswich, MA, USA). The RNA-Seq libraries were quantified using NEBNext^®^ Library Quant Kit for Illumina^®^ and pooled in equimolar amounts. Sequencing was performed on an Illumina Nova Seq 6000 platform with 150 bp paired end reads at the same core facility. Both long-read and short-read sequencing data were deposited in the NCBI Sequence Read Archive (SRA) under the accession number PRJNA1282630.

### 4.3. Analysis of RNA-Seq Data

ONT raw reads shorter than 500 bp and low-quality reads (quality score < 12) were removed, and the first 100 bp from the start of each read were trimmed using NanoFilt v2.7.1 [34]. The filtered reads from leaf and root samples were then used for *de novo* transcriptome assembly using RNA-Bloom v2.0.1 [35]. Illumina short reads were processed with Trimmomatic v0.39 [36] to remove adapter sequences and filter out low-quality reads. Replicates were pooled by tissue type, and tissue-specific transcriptomes were assembled using the Trinity *de novo* assembler v2.12.0 [37]. Assembled transcripts ≥ 300 bp (short reads) and 500 bp (long reads) were clustered using CD-HIT v4.6.8 [38] with a sequence identity threshold of 95%. To refine the reference transcriptomes, transcripts were filtered based on expression levels (TPM ≥ 2 in either leaf or root samples) by mapping short reads onto the clustered assemblies. The longest CDS were retained using a custom script (*longest_CDS.py*), and the final transcripts were confirmed via BLAST search against *C. chinensis* proteome. Filtered short reads (≥30 bp) were then mapped onto the final reference transcriptome using the Burrows Wheeler Alignment MEM algorithm v0.7.15 [39]. Differential expression between leaf and root tissues were assessed using DESeq2 v1.44.0 [40], applying a minimum fourfold change threshold. The obtained *p*-values were adjusted for multiple testing using the Benjamini–Hochberg (BH) FDR correction method at a significance level of α = 0.05 [41]. Functional enrichment analysis of DE transcripts was performed using the PANTHER Overrepresentation Test [42] and STRING v12.0 [43], and the results were summarized with REVIGO v1.20.0 to reduce redundancy among Gene Ontology (GO) terms [44].

### 4.4. Identification of Genes Involved in BIA Biosynthesis

To identify genes involved in BIA biosynthesis, the reference transcriptomes were searched against the proteomes of *Arabidopsis thaliana* (48,266 proteins) and *C. chinensis* (40,011 proteins). Candidate genes were translated into amino acid sequences and subjected to phylogenetic analysis with homologs from other species, including other *Coptis* species. A neighbor-joining (NJ) tree was constructed using BioEdit v7.2.5 [45], and the trees were visualized with Interactive Tree of Life (iTOL) v6 [46]. Multiple transcripts corresponding to the same gene were merged into one sequence based on phylogenetic results. The identified gene sequences have been deposited in NCBI GenBank under accession numbers PV842232 to PV842269.

### 4.5. Validation of RNA-Seq Results Using qRT-PCR

To validate the RNA-seq results, we employed qRT-PCR on 15 genes involved in BIA biosynthesis (Appendix A). RNA extraction was performed as described above, and first-strand cDNA was synthesized using the Maxima First Strand cDNA Synthesis Kit for RT-qPCR with dsDNase (Thermo Fisher Scientific, Waltham, MA, USA). qRT-PCR was conducted using the Luna^®^ Universal qPCR Master Mix (New England Biolabs, Ipswich, MA, USA) on a CFX96 Touch Real-Time PCR Detection System (Bio-Rad, Hercules, CA, USA). Each reaction was performed with three biological replicates of both leaf and root samples. The relative expression levels of target genes were calculated using the comparative Ct method (Applied Biosystems, Framingham, MA, USA), with 18S rRNA as the internal control.

### 4.6. Metabolite Profiling

#### 4.6.1. Targeted Analysis of Six BIA-Related Metabolites

A targeted analysis was conducted on six known BIA-related metabolites (Appendix A). Six metabolites (coptisine chloride, epiberberine chloride, columbamine chloride, palmatine chloride, jatrorrhizine chloride, and berberine chloride) were measured in the samples. Analytical standards, including coptisine chloride (Cat. No. SMB00314; purity > 98%), epiberberine chloride (Cat. No. TA9H93ED6E85; purity > 99%), columbamine chloride (Cat. No. PHL22604; purity > 98%), palmatine chloride (Cat. No. SMB00472; purity > 98%), jatrorrhizine chloride (Cat. No. PHL89530; purity > 98%), berberine chloride (Cat. No. 00900585; purity > 99%), and the internal standards, lidocaine (Cat. No. L1026; purity > 99%) and (1R)-(−)-10-camphorsulfonic acid (CA; Cat. No. 282146-25G; purity > 98%), were purchased from Sigma Chemical Co. (St. Louis, MO, USA).

To extract metabolites, dried tissues (10 mg) were homogenized with a metal ball in a screw-capped tube for 20 s at 1900 strokes/min on a GenoGrinder (Geno/Grinder 2000, SPEX SamplePrep., Metuchen, NJ, USA). Prior to extraction, the internal standard mixture (10 µM; 100 µM each of lidocaine and CA) was added to each sample. Then, metabolites were extracted following an established method [47]. Briefly, samples were extracted once in 1 mL of extraction solvent I (acetonitrile: isopropanol: water = 3:3:2 *v*/*v*/*v*) and twice with 0.5 mL of extraction solvent II (acetonitrile: water = 1:1 *v*/*v*) on a thermomixer (Thermomixer R, Eppendorf, Hamburg, Germany) with 1100 rpm at 4 °C for 5 min each time, followed by sonication for 15 min on ice and centrifugation at 13,000× *g* for 15 min at 4 °C. The resulting supernatant was transferred into a new 1.5 mL tube, lyophilized, and reconstituted in 100 µL of 50% methanol. Process blanks and pooled quality-control (QC) samples were prepared in parallel.

To proceed with MRM, the supernatants were combined and lyophilized, and the extracted compounds were analyzed using the triple quadrupole TSQ Altis mass spectrometers (Thermo Fisher Scientific, Waltham, MA, USA) coupled with the Vanquish Horizon UHPLC (Thermo Fisher Scientific, Waltham, MA, USA). The TSQ Altis was housed with a heated electrospray ionization (HESI) source using the following source settings: sheath gas flow of 50 Arb, auxiliary gas flow of 10 Arb, sweep gas flow of 1 Arb, and ion transfer tube temperature of 325 °C. The vaporizer temperature was set at 350 °C, and the spray voltage was 3.9 kV under the positive polarity. The scan time was set at 1 s, and the Q1 and Q3 resolutions of full width at half maximum (FWHM) were both 0.7. For the CID gas pressure, 1.5 mTorr was used. To determine the optimal fragments and collision energies (CES) for MRM transitions, we utilized the Xcalibur 4.1 software from Thermo Fisher Scientific for optimization. The selected fragment ions of each precursor with collision energies, including isotopically labeled precursors, are described in Appendix A. Alkaloid separation was performed using a Hypersil GOLD C18 column (100 mm × 2.1 mm, 1.9 µm) maintained at 35 °C. The mobile phase consisted of solvent A (0.1% formic acid in water) and solvent B (acetonitrile), with the following linear gradient: 25% B from 0 to 2 min; 25–75% B from 2 to 11 min; held at 75% B from 11 to 12.5 min; decreased to 25% B from 12.5 to 13.0 min; and held at 25% B for an additional 2 min. The flow rate was set to 0.3 mL/min, and the autosampler temperature was maintained at 4 °C. Two microliters of each sample were injected. Calibration curves generated from these standards (Appendix A) were used to quantify the amounts of the six targeted metabolites in *C. trifolia*.

#### 4.6.2. Untargeted Metabolomics

Untargeted metabolomics data were acquired using a Thermo Scientific Orbitrap Fusion Tribrid mass spectrometer (ThermoFisher Scientific, San Jose, CA, USA) interfaced with the Vanquish Horizon UHPLC system (ThermoFisher Scientific), which is equipped with a heated electron spray ionization (HESI) source. Samples prepared above were analyzed in a reversed-phase HSS T3 column (2.1 mm × 100 mm, 1.8 µm, Waters, Milford, MA, USA) at 40 °C. For positive mode, mobile phase A consisted of 0.1% formic acid in water (*v*/*v*), and mobile phase B consisted of 0.1% formic acid in 100% methanol (*v*/*v*). For negative ionization mode, mobile phase A contained 5 mM ammonium formate in water, while mobile phase B was methanol. The LC gradient was programmed to increase from 0% to 98% solvent B over 40 min. After sample injection, the system was held at 30% B for 22 min, ramped to 98% B over 8 min, maintained at 98% B for 1 min, then returned to 0% B in 1 min, followed by an 8 min re-equilibration at 0% B. Flow rate was 0.4 mL/min with a 10 µL injection. Autosampler temperature was maintained at 4 °C throughout the analyses. MS/MS parameters were as follows: spray voltage +3.5 kV (positive) or −2.5 kV (negative); sheath gas 40 arbitrary units (AU); auxiliary gas 8 AU; sweep gas 1 AU; capillary temperature 275 °C; auxiliary gas heater 320 °C; S-Lens RF level 35%. Full MS1 scans were acquired in the Orbitrap at a resolution of 120,000 (*m*/*z* 200) over a scan range of *m*/*z* 60–900, with an automatic gain control (AGC) target of 1 × 10^6^ and maximum injection time of 100 ms. Data-dependent MS/MS (2 s cycle time) was performed with quadrupole isolation (1.0 *m*/*z*), stepped higher-energy collisional dissociation (HCD, 20/35/50%), and Orbitrap detection at a resolution of 60,000. Dynamic exclusion was set to 2.5 s.

#### 4.6.3. Quality Control and Method Validation

The method was validated in accordance with international guidelines, evaluating selectivity, identification capacity, limits of detection and quantification, recovery, carryover, and matrix effects, as well as intra- and inter-assay accuracy and precision, prior to its application to 66 authentic compounds [48] (Appendix A). To monitor analytical reproducibility, a panel of 66 standard compounds was analyzed at the beginning and end of each batch to verify retention time stability, mass accuracy (<5 ppm), and overall system performance, while blank injections were included to assess potential carryover.

#### 4.6.4. Data Processing, Compound Detection, Grouping, Alignment, and Background Subtraction

Raw LC–MS/MS data files were processed using Compound Discoverer software (version 3.3, Thermo Fisher Scientific, Waltham, MA, USA). The untargeted metabolomics workflow was performed according to the standard pipeline recommended by the vendor, with adjustments for study-specific requirements. Data were first recalibrated using background signals and two animal compounds and nine isotopically labeled internal standards to ensure retention time stability across all UHPLC separations and maintain mass accuracy within <5 ppm. Peak detection was performed with an intensity threshold of 0.5 × 10^6^ and a minimum signal-to-noise ratio of 3. Retention time alignment across all samples was carried out using a mass tolerance of 5 ppm and a retention time window of 0.2 min. Grouped compounds based on accurate mass (±5 ppm), isotope pattern matching, and adduct formation rules (H^+^, NH_4_^+^, and formate) were selected, and compounds detected in blank samples at comparable intensity levels were excluded to remove background contaminants. Peak intensities were normalized using internal standard responses and further corrected with QC-based normalization (pooled QC samples injected every 8–10 runs). Features with a coefficient of variation (%CV) > 30% in pooled QC samples were excluded from downstream analysis. Annotation was achieved by matching accurate mass and MS/MS spectra against multiple spectral libraries integrated within Compound Discoverer, including mzCloud, ChemSpider, and KEGG.

#### 4.6.5. Metabolite Data Analysis

For untargeted metabolomics data, statistical analysis was performed using the MetaboAnalyst 6.0 web platform (www.metaboanalyst.ca; assessed on 16 May 2025). The raw intensities of metabolites were log_10_-transformed and auto-scaled (mean-centered and divided by the standard deviation of each variable) to normalize variance across metabolites. Principal Component Analysis (PCA) was conducted as an unsupervised method to assess overall variance in the dataset and to visualize the clustering patterns of samples and potential outliers. Partial Least Squares Discriminant Analysis (PLS-DA) was then applied as a supervised method to maximize separation between two groups and to identify metabolites contributing most strongly to group discrimination. To identify differentially accumulated metabolites, two-tailed Student’s *t*-tests were performed. The *p*-values were adjusted for false discovery rate (FDR) using the BH method [41]. Metabolites with an adjusted *p*-value < 0.05 and a two-fold change were considered statistically significant. All data processing steps, statistical tests, and visualizations (including PCA and PLS-DA score plots) were completed within the MetaboAnalyst platform. Lastly, six targeted metabolites were compared between leaf and root samples using a two-tailed Student’s *t*-test with a *p*-value < 0.05.

## 5. Conclusions

In this study, we generated a high-quality reference transcriptome for *C. trifolia* by integrating ONT and Illumina sequencing technologies. Our refined transcriptome, supported by rigorous filtering and validation, enabled accurate detection of gene expression patterns and functional annotation. Comparative transcriptomic and metabolomic analyses revealed that there was higher accumulation of some BIAs, such as epiberberine, berberine, columbamine, and coptisine, as well as more abundant expression of biosynthetic genes in roots compared to leaves. These results demonstrate that *C. trifolia*, belonging to a sister clade to Asian medicinal *Coptis* species, shares conserved and functionally relevant BIA biosynthetic pathways. Phylogenetic and expression analyses suggest that *C. trifolia* retains multifunctional *CAS* homologs that can catalyze various steps in the BIA pathway, similar to other *Coptis* species. Despite experiencing a single round of whole-genome duplication, multifunctionality of *CAS* persists within a single clade, in contrast to the duplicated and functionally divergent *CFS* and *SPS* genes in Papaveraceae. Furthermore, transcription factors *bHLH1* and *WRKY1*, previously shown to regulate BIA biosynthesis in *C. chinensis* and *C. japonica*, were also identified in *C. trifolia*, with *bHLH1* showing higher expression and thus likely exerting a more significant regulatory influence. Although our results strongly suggest the functions of key candidate genes, such as *Ctr_SPS* or *Ctr*_*bHLH1*, functional characterization has not yet been performed. Future studies employing heterologous expression, enzyme assays, and gene-editing approaches will be essential to validate the roles of these genes and clarify lineage-specific multifunctionality within the BIA pathway. Such efforts will deepen our understanding of regulatory mechanisms, guide metabolic engineering strategies, and strengthen the evolutionary framework for BIA biosynthesis across *Coptis* species. Together, our findings establish *C. trifolia* as a promising comparative system for studying BIA biosynthesis and evolution in Ranunculaceae. This study provides a valuable genomic resource for future functional studies, metabolic engineering, and conservation efforts targeting this medicinal species. Further experimental validation of candidate genes and transcriptional regulators will deepen our understanding of BIA pathway regulation and its evolutionary dynamics across the *Coptis* genus.

## Figures and Tables

**Figure 1 ijms-26-09704-f001:**
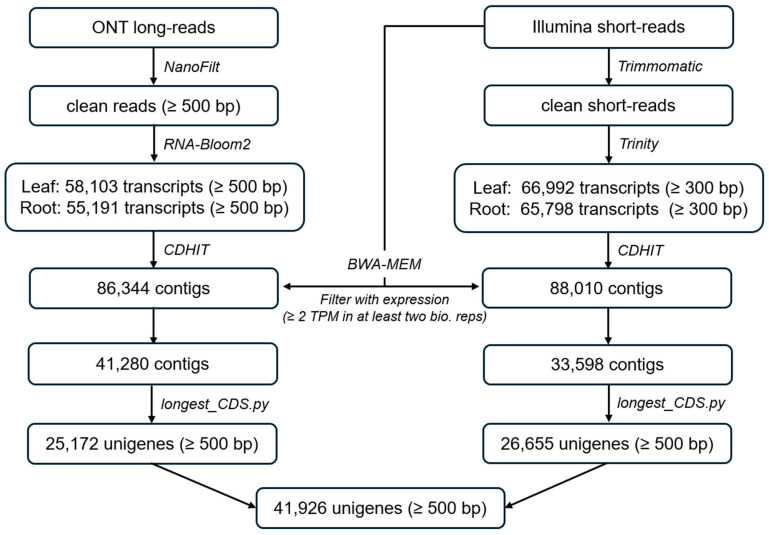
Construction of the reference transcriptome using long-read and short-read sequencing.

**Figure 2 ijms-26-09704-f002:**
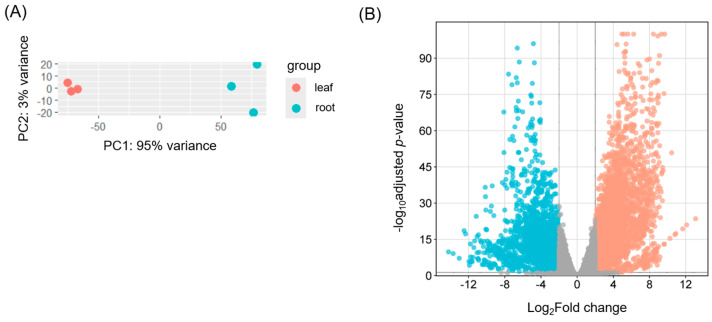
Transcriptome profiles in leaf and root tissues. (**A**) Principal Component Analysis (PCA) score plot of transcriptome profiles. (**B**) Volcano plot of differentially expressed transcripts. Orange and cyan dots indicate transcripts more abundant in leaf and root tissues, respectively, compared to the other tissue. Gray dots indicate no differential expression between the two tissues. The cutoff criteria were set at |log_2_ fold change| ≥ 2 with an adjusted *p*-value < 0.05.

**Figure 3 ijms-26-09704-f003:**
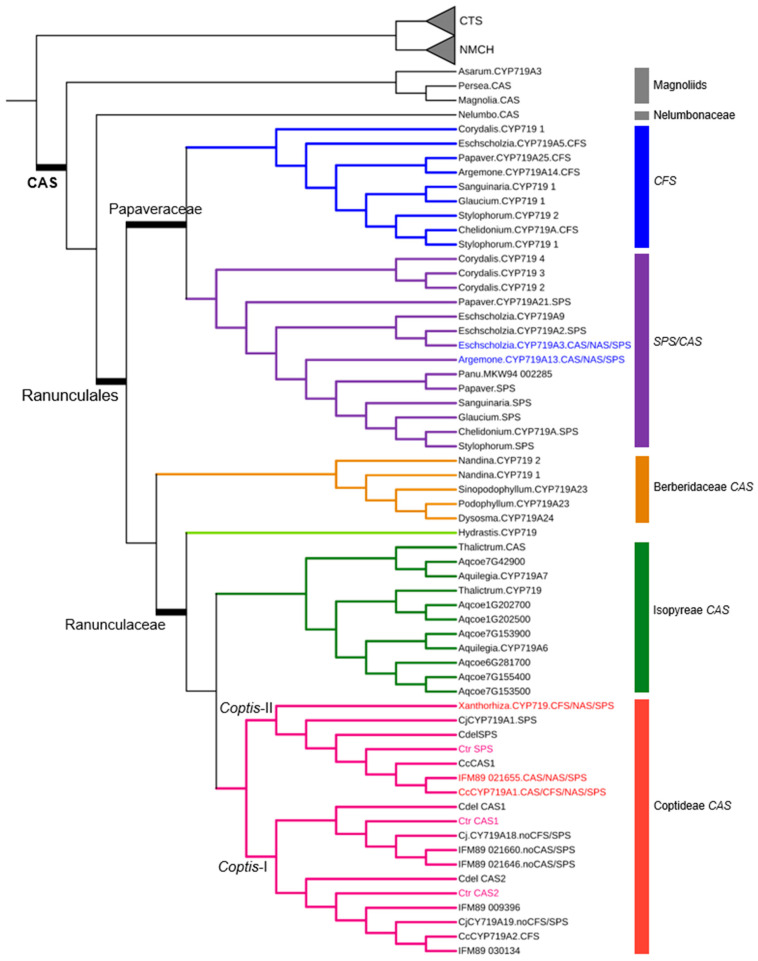
Neighbor-Joining (NJ) tree of *CAS* gene homologs. *CTS* and *NMCH* sequences were used as outgroups. *CAS* gene homologs from *C. trifolia* are shown in pink. In the phylogenetic tree, blue and purple lines represent *CFS* and *SPS/CAS* of Papaveraceae, respectively, while orange lines indicate *CAS* of Berberidaceae. Light green, green, and magenta lines correspond to the Hydrastis *CYP719* sequence, and *CAS* of Isopyreae and Coptideae, respectively. Genes with multifunctional CAS/NAS/SPS or CAS/CFS/NAS/SPS activities are highlighted in blue and red, respectively. The information of sequences used in tree construction are provided in Appendix A.

**Figure 4 ijms-26-09704-f004:**
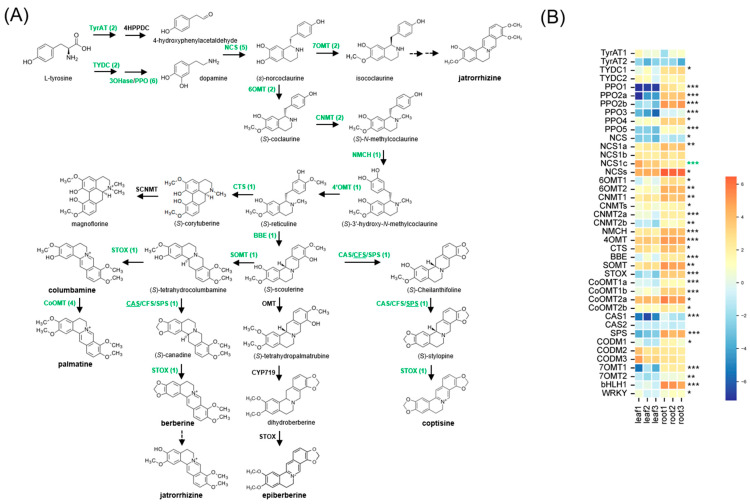
Putative biosynthetic pathways of BIA in *Coptis* species (**A**) and expression patterns of putative BIA biosynthetic genes (**B**). In the pathway diagram, enzymes identified from *C. trifolia* are shown in green text, with the number of gene homologs indicated in parentheses. Metabolites targeted for profiling are highlighted in bold. Underlined genes are putatively responsible for catalyzing the reaction. In the heatmap, gene expression values are log_2_-transformed TPM. Asterisks indicate significantly higher gene expression in roots (black) or leaves (green), with significance levels denoted as adjusted *p* < 0.05 (*), *p* < 0.01 (**), and *p* < 0.001 (***).

**Figure 5 ijms-26-09704-f005:**
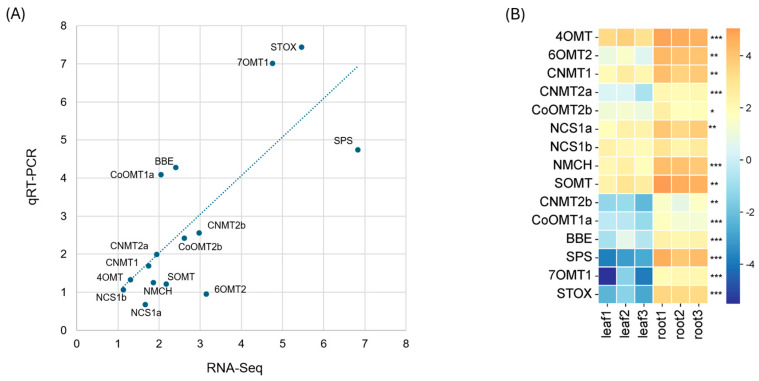
Validation of RNA-Seq results using 15 selected BIA biosynthetic genes. (**A**) Correlation between RNA-Seq and qRT-PCR results based on log_2_-fold changes between leaf and root samples. (**B**) Heatmap of RNA-Seq data showing log_2_-transformed TPM for selected genes. Asterisks indicate significantly higher gene expression in roots than leaves, with significance levels denoted as adjusted *p* < 0.05 (*), *p* < 0.01 (**), and *p* < 0.001 (***).

**Figure 6 ijms-26-09704-f006:**
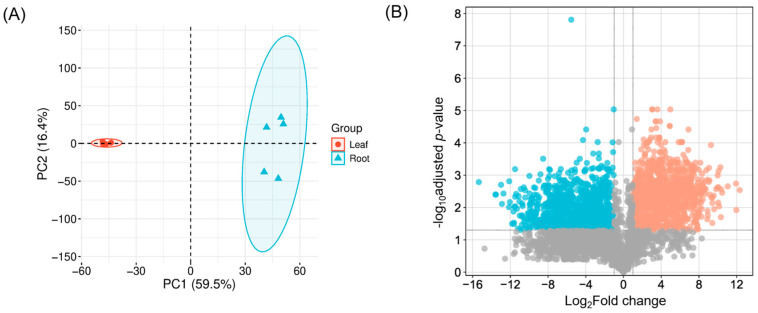
Metabolite profiling. (**A**) Score plot of Principal Component Analysis (PCA) of untargeted metabolites. (**B**) Volcano plot of differentially accumulated metabolites. Orange and cyan dots indicate metabolites more abundant in leaf and root tissues, respectively, compared to the other tissue. Gray dots indicate no differential expression between the two tissues. The cutoff criteria were set at |log_2_ fold change| ≥ 1 with an adjusted *p*-value < 0.05.

**Figure 7 ijms-26-09704-f007:**
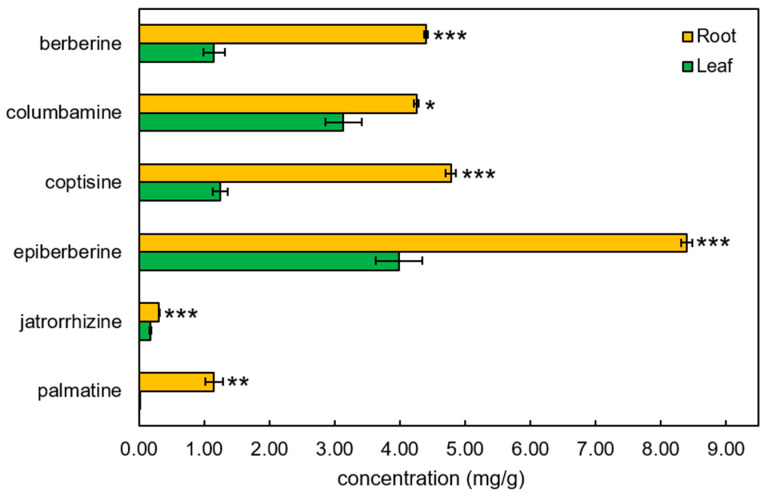
Contents of six BIAs in leaf and root tissues of *C. trifolia*. Differentially accumulated BIAs between leaf and root samples are indicated by asterisks: *p* < 0.05 (*), *p* < 0.01 (**), and *p* < 0.001 (***) from Student’s *t*-test. The bar within the box represents the mean ± standard error.

## Data Availability

Data is contained within the article or Appendix A.

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
