# Peer review of "Transcriptomic and Metabolomic Insights into Benzylisoquinoline Alkaloid Biosynthesis in Goldthread (*Coptis trifolia*)"

_ijms, 2025, doi:10.3390/ijms26199704_

Round 1

Reviewer 1 Report

Comments and Suggestions for Authors

Dear Author: 

This manuscript presents a comprehensive transcriptomic and metabolomic analysis of Coptis trifolia, a North American species with medicinal potential, focusing on the biosynthesis of benzylisoquinoline alkaloids (BIAs). The study leverages both long-read (ONT) and short-read (Illumina) sequencing technologies to assemble a high-quality transcriptome and identify key genes involved in BIA pathways. The work is well-structured, methodologically sound, and provides valuable genomic resources for a non-model plant species. The findings contribute to our understanding of BIA biosynthesis evolution and regulation within the Ranunculaceae family.

Here are the revision suggestions for this manuscript:
(1) In the "Materials and Methods" section, the description of biological replicate samples should be more detailed. The manuscript states that there were three biological replicates for the transcriptome study and five for the metabolome study. However, it is unclear whether the ONT sequencing was performed on pooled samples or individual replicates.
(2) In the discussion or conclusion section, prospects for follow-up functional validation should be included. Although phylogenetic and expression analyses strongly suggest the functions of these genes, functional validation of key candidate genes (e.g., Ctr_SPS, Ctr_CNMT1) has not been completed. Please consider conducting in-depth functional studies on these candidate genes in subsequent research and emphasize the follow-up validation results in future manuscripts.
(3) For metabolomics, the use of an FDR-adjusted *p*-value of less than 0.1 as the criterion for statistical significance is relatively lenient. It would be better if a reasonable justification for this threshold could be provided, or if a stricter cutoff (e.g., FDR < 0.05) could be adopted.
(4) The manuscript occasionally uses the terms "transcripts" and "genes" interchangeably. Maintaining consistency in the use of these terms would improve the readability of the article.

The final decision is to accept with minor revisions. Please ask the authors to address the above points in detail.

Date: September 12, 2025

Author Response

We sincerely thank you for taking the time to provide valuable feedback on our manuscript. Your thoughtful comments and suggestions have significantly improved the quality and clarity of our work.

We have carefully addressed each of your points and revised the manuscript accordingly. A detailed response to your comments has been provided, with changes highlighted in red text and additional explanations included where necessary.

We greatly appreciate your efforts in helping us refine our work and hope that the revisions meet your expectations. Please do not hesitate to reach out if further clarifications are needed.

Thank you again for your insights and contributions.

(1) In the "Materials and Methods" section, the description of biological replicate samples should be more detailed. The manuscript states that there were three biological replicates for the transcriptome study and five for the metabolome study. However, it is unclear whether the ONT sequencing was performed on pooled samples or individual replicates.

We provided sampling information on the ONT sequencing in lines 451 – 452: “For ONT sequencing, equal amounts of RNA from three leaf and three root samples were pooled”.

(2) In the discussion or conclusion section, prospects for follow-up functional validation should be included. Although phylogenetic and expression analyses strongly suggest the functions of these genes, functional validation of key candidate genes (e.g., Ctr_SPS, Ctr_CNMT1) has not been completed. Please consider conducting in-depth functional studies on these candidate genes in subsequent research and emphasize the follow-up validation results in future manuscripts.

Thank you for your valuable comments. We have incorporated your suggestions into the Conclusions section as follows.

Lines 643-649: Although our results strongly suggest the functions of key candidate genes, such as Ctr_SPS or Ctr_bHLH1, functional characterization has not yet been performed. Future studies employing heterologous expression, enzyme assays, and gene-editing approaches will be essential to validate the roles of these genes and clarify lineage-specific multifunctionality within the BIA pathway. Such efforts will deepen our understanding of regulatory mechanisms, guide metabolic engineering strategies, and strengthen the evolutionary framework for BIA biosynthesis across Coptis species.

(3) For metabolomics, the use of an FDR-adjusted *p*-value of less than 0.1 as the criterion for statistical significance is relatively lenient. It would be better if a reasonable justification for this threshold could be provided, or if a stricter cutoff (e.g., FDR < 0.05) could be adopted.

We applied for an FDR-adjusted p-value of less than 0.05 (line 621), which reduced DE metabolites (2226; 978 in leaves, 1248 in roots) (Lines 248-249). In addition, Figure 6 was updated with an adjusted p-value.

(4) The manuscript occasionally uses the terms "transcripts" and "genes" interchangeably. Maintaining consistency in the use of these terms would improve the readability of the article.

We tried to keep consistency in using these terms, for example, we used “transcripts” when we refer to the products of transcription. In other words, we used “genes”.

Reviewer 2 Report

Comments and Suggestions for Authors

The author constructed the gene network of Benzylisoquinoline Alkaloid Biosynthesis in Goldthread through the combined analysis of metabolomics and transcriptomics. The research findings can provide reference information for deciphering the genetics of Goldthread.

  1. The title of manuscript does not quite match the content.
  2. Supplementary figures are mentioned in the main text but are not found in the attachments, such as Figure S6. This result is crucial for verifying the reliability of the transcriptomic data. It is suggested to place it in the main text and indicate whether the trend of the transcriptomic data is consistent with the qPCR results. If other attachments have indeed not been uploaded, it is recommended to complete them.
  3. It is recommended to supplement the content of statistical analysis methods in the Materials and Methods section or where there is statistical analysis in figures and tables.
  4. The criteria for sample selection should be detailed in the Materials and Methods section. For example, the sampling period, tissue parts, and sample quantity should be specified.
  5. The established biosynthetic pathways of Benzylisoquinoline Alkaloid Biosynthesis should be described in the Introduction section.
  6. The evidence for the conclusion that “Conversely, Coptis genes in the Coptis-II clade exhibit multifunctional enzyme activities, including CAS, CFS, SPS, and (S)-nandinine synthase (NAS) activities (Figure 3)” is not sufficient.

Author Response

We sincerely thank you for taking the time to provide valuable feedback on our manuscript. Your thoughtful comments and suggestions have significantly improved the quality and clarity of our work.

We have carefully addressed each of your points and revised the manuscript accordingly. A detailed response to your comments has been provided, with changes highlighted in red text and additional explanations included where necessary.

We greatly appreciate your efforts in helping us refine our work and hope that the revisions meet your expectations. Please do not hesitate to reach out if further clarifications are needed.

Thank you again for your insights and contributions.

The author constructed the gene network of Benzylisoquinoline Alkaloid Biosynthesis in Goldthread through the combined analysis of metabolomics and transcriptomics. The research findings can provide reference information for deciphering the genetics of Goldthread.

  1. The title of manuscript does not quite match the content.

We have revised the title to: “Transcriptomic and Metabolomic Insights into Benzylisoquinoline Alkaloid Biosynthesis in Goldthread (Coptis trifolia)”

  1. Supplementary figures are mentioned in the main text but are not found in the attachments, such as Figure S6. This result is crucial for verifying the reliability of the transcriptomic data. It is suggested to place it in the main text and indicate whether the trend of the transcriptomic data is consistent with the qPCR results. If other attachments have indeed not been uploaded, it is recommended to complete them.

Thank you for this recommendation. We moved Figure S6 to the main text as Figure 5.

  1. It is recommended to supplement the content of statistical analysis methods in the Materials and Methods section or where there is statistical analysis in figures and tables.

We included statistical methods as below.

  • Lines: 477-481 – statistical test for transcriptome data
  • Lines: 618-621, 623 - 625 - statistical test for metabolite data
  • Statistical method was added in Figure 7 as it was not described in Methods.
  • Line 280: from Student’s t-test. The bar within the box represents the mean ± standard error.

  1. The criteria for sample selection should be detailed in the Materials and Methods section. For example, the sampling period, tissue parts, and sample quantity should be specified.

Thank you for your valuable comments. Because this species is perennial and collected from a wild population, it is difficult to accurately estimate its age. Therefore, we collected materials from newly emerged plants. To further reduce biological variation, we selected two individual plants with similar morphology per replicate. We have revised the Materials section accordingly, as shown below.

Lines 438-441: One year after transplantation, we tagged newly emerged leaves from rhizome. Then, mature leaf with its roots, including rhizome, was collected for each biological replicate as shown in Figure S7. Two individuals with similar morphology were included in each replicate, as the age of this perennial species cannot be determined.

  1. The established biosynthetic pathways of Benzylisoquinoline Alkaloid Biosynthesis should be described in the Introduction section.

We included the established biosynthetic pathways of BIA in the Introduction as below.

Lines 60 – 66: BIA originates from L-tyrosine. Dopamine and 4-hydroxyphenylacetaldehyde condense via norcoclaurine synthase (NCS) to yield (S)-norcoclaurine, the first pathway-specific intermediate [9-11]. Through the canonical sequence, (S)-norcoclaurine is converted to the key intermediate (S)-reticuline, with (S)-coclaurine as an intermediate and branch point. Then, (S)-reticuline is converted to (S)-scoulerine by the berberine bridge enzyme (BBE), followed by CYP719A-mediated steps, O-methyltransferases, and final oxidation, which produces epiberberine, berberine, coptisine, and columbamine [9-11].

  1. The evidence for the conclusion that “Conversely, Coptis genes in the Coptis-II clade exhibit multifunctional enzyme activities, including CAS, CFS, SPS, and (S)-nandinine synthase (NAS) activities (Figure 3)” is not sufficient.

We have revised this sentence as follows to better clarify our intention.

Lines 197 – 200: Conversely, Coptis genes in the Coptis-II clade are suggested, based on previous functional studies, to exhibit multifunctional enzyme activities, including CAS, CFS, SPS, and (S)-nandinine synthase (NAS) activities [11,26,27] (Figure 3), although additional validation is required. 

Reviewer 3 Report

Comments and Suggestions for Authors

The manuscript by Koh et al., presents a comprehensive transcriptomic and metabolomic analysis of Coptis trifolia, highlighting genes involved in benzylisoquinoline alkaloid (BIA) biosynthesis and comparing patterns of accumulation between leaf and root tissues. The authors analyzed long-read and short-read sequencing and coupled with metabolite profiling, providing valuable insights and resources for future studies on this medicinally relevant genus. The study is timely and contributes to the broader understanding of secondary metabolism in Ranunculaceae. Overall, the manuscript is well written, and the results are clearly presented. However, some areas in this manuscript require clarification, expansion, and additional data to strengthen the conclusions.

The most critical concern that I have is sampling. Although the authors stated in the method section that “One year after transplantation, two leaves and three roots, including rhizomes, were collected for each biological replicate”, there are still high chance of variation in the sample collection. What leaf stage were those two leaves at, and the age of the individual plants used in this study? Since leaf and root architecture are large and complex, I strongly suggest the authors to include the specific details of what leaf and root tissues were collected.

The authors identified two transcription factors, bHLH1 and WRKY1, that regulate BIA biosynthesis in Coptis species. I wonder whether the authors could take a look or at least discuss the potential transcriptional cascade/networks these two TFs lead in term of the DE that were identified in this study. Since the consensus DNA binding motifs of these two TFs are well studied, it would be very resourceful if the authors could hint or emphasize this regulation.  

Only six BIAs were quantified in targeted metabolomics. I wonder whether it is because other pathway intermediates were not quite detectable. If possible (not required for the scope of this manuscript of course), including intermediate compounds would help connect transcriptomic regulation with metabolic flux.

Fore Fig. 2B, I suggest the authors to state the cutoff criteria for p value and fold change in the figure captions. Also, the dot legend in the volcano plot is missing in this panel. The same issue is present in Fig. 5.

Author Response

We sincerely thank you for taking the time to provide valuable feedback on our manuscript. Your thoughtful comments and suggestions have significantly improved the quality and clarity of our work.

We have carefully addressed each of your points and revised the manuscript accordingly. A detailed response to your comments has been provided, with changes highlighted in red text and additional explanations included where necessary.

We greatly appreciate your efforts in helping us refine our work and hope that the revisions meet your expectations. Please do not hesitate to reach out if further clarifications are needed.

Thank you again for your insights and contributions.

The manuscript by Koh et al., presents a comprehensive transcriptomic and metabolomic analysis of Coptis trifolia, highlighting genes involved in benzylisoquinoline alkaloid (BIA) biosynthesis and comparing patterns of accumulation between leaf and root tissues. The authors analyzed long-read and short-read sequencing and coupled with metabolite profiling, providing valuable insights and resources for future studies on this medicinally relevant genus. The study is timely and contributes to the broader understanding of secondary metabolism in Ranunculaceae. Overall, the manuscript is well written, and the results are clearly presented. However, some areas in this manuscript require clarification, expansion, and additional data to strengthen the conclusions.

The most critical concern that I have is sampling. Although the authors stated in the method section that “One year after transplantation, two leaves and three roots, including rhizomes, were collected for each biological replicate”, there are still high chance of variation in the sample collection. What leaf stage were those two leaves at, and the age of the individual plants used in this study? Since leaf and root architecture are large and complex, I strongly suggest the authors to include the specific details of what leaf and root tissues were collected.

Thank you for your valuable comments. Because this species is perennial and collected from a wild population, it is difficult to accurately estimate its age. Therefore, we collected materials from newly emerged plants. To further reduce biological variation, we selected two individual plants with similar morphology per replicate. We have revised the Materials section accordingly, as shown below.

Lines 438-441: One year after transplantation, we tagged newly emerged leaves from rhizome. Then, mature leaf with its roots, including rhizome, was collected for each biological replicate as shown in Figure S7. Two individuals with similar morphology were included in each replicate, as the age of this perennial species cannot be determined.

 The authors identified two transcription factors, bHLH1 and WRKY1, that regulate BIA biosynthesis in Coptis species. I wonder whether the authors could take a look or at least discuss the potential transcriptional cascade/networks these two TFs lead in term of the DE that were identified in this study. Since the consensus DNA binding motifs of these two TFs are well studied, it would be very resourceful if the authors could hint or emphasize this regulation.  

We appreciate this suggestion. DNA-binding preferences of bHLH1 and WRKY1 in Coptis (E/G-box and W-box, respectively) have been characterized, and several direct TF–target interactions have been reported in C. japonica and C. chinensis; however, a comprehensive, system-level cascade remains incomplete [19–21]. Our study quantifies transcript abundances from de novo transcriptomes, and—without a chromosome-level genome—the assembled sequences do not include upstream promoter regions, precluding promoter-motif enrichment and in silico target prediction at this time. We have added text to the Discussion prioritizing future validation to define bHLH1/WRKY1-centered regulatory cascades in C. trifolia in lines 429 – 432.

 Only six BIAs were quantified in targeted metabolomics. I wonder whether it is because other pathway intermediates were not quite detectable. If possible (not required for the scope of this manuscript of course), including intermediate compounds would help connect transcriptomic regulation with metabolic flux.

Unfortunately, untargeted metabolomics detected only one metabolite (berberine) in the BIA pathway. Although intermediate compounds would have provided stronger links between transcriptomic regulation and metabolic flux, such analyses are beyond the scope of this manuscript.

Fore Fig. 2B, I suggest the authors to state the cutoff criteria for p value and fold change in the figure captions. Also, the dot legend in the volcano plot is missing in this panel. The same issue is present in Fig. 5.

Figure 2B and Figure 6B, the cutoff criteria were added. As for Figure 6 (previously Fig. 5), the dot legend was described in Figure legend.

Round 2

Reviewer 2 Report

Comments and Suggestions for Authors

Accept in present form